

# Which are important soil parameters influencing the spatial heterogeneity of [14]C in soil organic matter?

Stephan John[1*], Gerrit Angst[2], Kristina Kirfel[3], Sebastian Preußer[4], Carsten W. Mueller[2], Christoph Leuschner[3], Ellen Kandeler[4], Janet Rethemeyer[1]

[1] Institute for Geology and Mineralogy, University of Cologne, Zülpicher Straße 49a, 50674 Cologne, Germany

[2] Lehrstuhl für Bodenkunde, TU München, Emil-Ramann-Straße 2, 85354 Freising, Germany

[3] Albrecht von Haller Institute for Plant Sciences, Georg-August-University Göttingen, Untere Karspüle 2, 37073 Göttingen, Germany

[4] Institute of Soil Science and Land Evaluation, Soil Biology Section, University of Hohenheim, Emil-Wolff-Straße 27, 70593 Stuttgart, Germany

Correspondence: Stephan John, E-mail: sjohn1@uni-koeln.de, Tel.: +49 0221 470 7318

**Abstract**

Radiocarbon ([14]C) analysis is an important tool that can provide information on the dynamics of organic matter in soils. Radiocarbon concentrations of soil organic matter (SOM) however, reflect the heterogeneous mixture of various organic compounds and are affected by different chemical, biological, and physical soil parameters. These parameters can vary strongly in soil profiles and thus affect the spatial distribution of the apparent [14]C age of SOM considerably. The heterogeneity of SOM and its [14]C signature may be even larger in subsoil horizons, which are thought to receive organic carbon inputs following preferential pathways. This will bias conclusions drawn from [14]C analyses of individual soil profiles considerably. We thus investigated important soil parameters, which may influence the [14]C distribution of SOM as well as the spatial heterogeneity of [14]C distributions in soil profiles. The suspected strong heterogeneity and spatial variability, respectively of bulk SOM is confirmed by the variable [14]C distribution in three 185 cm deep profiles in a Dystric Cambisol. The [14]C contents are most



variable in the C horizons because of large differences in the abundance of roots there. The
distribution of root biomass and necromass and its organic carbon input is the most important
factor affecting the $^{14}$C distribution of bulk SOM. The distance of the soil profiles to a beech
did not influence the horizontal and vertical distribution of roots and $^{14}$C concentrations. Other
parameters were found to be of minor importance including microbial biomass-derived carbon
and soil texture. The microbial biomass however, may promote a faster turnover of SOM at hot
spots resulting in lower $^{14}$C concentration there. Soil texture had no statistically significant
influence on the spatial $^{14}$C distribution of bulk SOM. However, SOM in fine silt and clay sized
particles (<6.3 µm) yields slightly higher $^{14}$C concentrations than bulk SOM particularly at
greater soil depth, which is in contrast to previous studies where silt and clay fractions contained
older SOM stabilized by organo-mineral interaction. $^{14}$C contents of fine silt and clay correlate
with the microbial biomass-derived carbon suggesting a considerable contribution of microbial-
derived organic carbon. In conclusion, $^{14}$C analyses of bulk SOM mainly reflect the spatial
distribution of roots, which is strongly variable even on a small spatial scale of few meters. This
finding should be considered when using $^{14}$C analysis to determine SOM.

**Keywords:** organic matter heterogeneity, radiocarbon, soil carbon dynamic, subsoils

**1. Introduction**
Radiocarbon analysis is a helpful tool to determine the dynamics of organic matter in soils as it
provides a direct measure of the time elapsed since atmospheric $CO_2$ was fixed by plants
through photosynthesis (Trumbore, 2009). However, soil organic matter (SOM) is a complex
mixture of organic components derived from different sources at various stages of
decomposition (Rethemeyer et al., 2004; Trumbore and Zheng, 1996). Consequently, $^{14}$C
concentrations of SOM reflect the average composition and apparent mean residence time
(MRT), respectively of a wide range of compounds turning over on different time scales. The
$^{14}$C content of SOM is affected by various soil parameters most importantly by the input of
carbon from plant litter and roots. Other important factors influencing SOM dynamics and thus
$^{14}$C contents include physical parameters such as soil texture, various chemical parameters like
pH, moisture, nutrients, and biological factors such as the presence of roots and the
microorganisms. These factors can vary strongly in soil horizons (Don et al., 2007; Enowashu
et al., 2009; Kramer et al., 2013; Schöning et al., 2006b) which is supposed to result in a
significant spatial variability in $^{14}$C contents of SOM.





The heterogeneity of SOM is suggested to increase with increasing soil depth. In contrast to
surface soils where the input of organic carbon (OC) derived mainly from fresh plant litter,
subsoil horizons receive OC mainly from root biomass (Rasse et al., 2005), dissolved organic
matter (Kaiser and Guggenberger, 2000), and particulate organic matter transported downward
by physical and/or biological processes (Don et al., 2008). The transport of the OC into deeper
horizons was found to follow preferential flow paths like root channels and animal burrows
(Bundt et al., 2001; Chabbi et al., 2009; Don et al., 2008), which results in a considerable spatial
heterogeneity. Thus $^{14}$C contents in subsoil are supposed to vary much stronger on a small
spatial scale compared to topsoil. Because of the expense of $^{14}$C analysis this has not yet been
investigated and most studies rely on $^{14}$C analyses of single soil profiles (e.g. Eusterhues et al.,
2005, 2007; Rumpel et al., 2002, 2004).
Beside a stronger spatial heterogeneity, the turnover of SOM is reduced considerably at greater
soil depth as suggested by strongly decreasing $^{14}$C concentrations in subsoil horizons (e.g.
Eusterhues et al., 2005, 2007; Rumpel and Kögel-Knabner, 2011; Rumpel et al., 2002, 2004;
Torn et al., 1997). Until now it is not well understood if the low $^{14}$C concentrations in subsoils
reflect the accumulation of chemically more refractory organic compounds (Eusterhues et al.,
2007), the stabilization of SOM by organo-mineral interaction (Salomé et al., 2010; Schöning
and Kögel-Knabner, 2006) or a lower abundance of microbial biomass and a resulting reduced
SOM turnover (Fierer et al., 2003; Salomé et al. 2010). Several previous studies suggest that in
subsoils the interaction of SOM with the mineral soil matrix is the most important process
controlling the increase in the apparent $^{14}$C age of SOM with depth rather than the accumulation
of degradation resistant compounds. For example Eusterhues et al. (2005) and Mikutta et al.
(2006) have shown that OC adsorbed to iron and aluminium oxides and/or clay minerals is
several hundred to thousand years older than bulk OC. However, results of Fontaine et al.
(2007) suggest that the content of such soil minerals increases only little with depth, which
could not explain the large shift in MRT from years to several thousand years (in 0-20 vs. 60-
80 cm) observed in this study. The slow turnover of OC thus was assumed to be a result of the
significant reduction of the microbial biomass at greater depth. This was also shown by Fierer
et al. (2003) who found that the microbial communities inhabiting deeper soil horizons are more
carbon limited than those in the surface soil. Thus, the low abundance of the microbial biomass
and the lack of fresh substrate may significantly reduce OC turnover in deeper soil horizons
promoting high apparent $^{14}$C ages. As roots were found to introduce relatively fresh OC into
deeper soil horizons, the substrate limitation and the associated slow OC turnover is supposed
to be absent near roots resulting in relatively young apparent $^{14}$C ages (Chabbi et al., 2009).



Our study was designed to clarify the driving factors for the spatial heterogeneity of $^{14}$C
contents of the organic matter in subsoils. Two main aspects were investigated including a) the
analysis of different soil parameters that may affect the $^{14}$C distribution in subsoil profiles, and
b) the spatial heterogeneity of SOM along a 3.15 m long transect increasing in distance to a
beech. The latter makes it possible to investigate the spatial effect of the vegetation, most
importantly the distribution of roots, on the distribution of OC and its $^{14}$C content. The analysed
soil parameters, which may have a significant influence on the $^{14}$C depth distribution include
biological (root biomass, microbial biomass), chemical (OC and N content, C/N ratio), and
physical variables (particle size distribution), which were measured in three profiles along the
transect. The influence of the different soil parameter given above on the $^{14}$C distribution of
SOM was evaluated using principle component analysis (PCA).

**2. Material and Methods**

**2.1 Site description and sampling**
The study site is located in the Grinderwald in Northern Germany about 40 km north-west of
Hannover (52°34'22.115 N, 9°18'49.762 E). The beech forest (*Fagus sylvatica L.*) was
established in 1916. The mean annual precipitation is 762 mm and the mean temperature in the
period 1981 - 2010 was 9.7 °C measured by the German meteorological service monitoring
station (station Nienburg). The soil is classified as Dystric Cambisol (IUSS Working Group
WRB, 2014) developed on Pleistocene (Saale Glacial) melt-water deposits (Jordan, 1980) with
an acidic pH (3.4 - 4.5) and a mainly sandy texture (77.3 % sand, 18.4 % silt, and 4.4 % clay).
Three profiles were sampled on a 3.15 m long and 1.85 m deep transect increasing in distance
to a beech (profile A: 0 cm, D: 135 cm, and G: 270 cm). Samples were taken from seven soil
depths (10, 35, 60, 85, 110, 135, and 160 cm) below the A horizon (Fig. 1). All samples, with
the exception of samples used for density and particle size fractionation, were sieved <2 mm
and freeze-dried prior to analysis.

**2.2 Chemical parameters**
Carbon and nitrogen contents were analyzed by dry combustion using elemental analyzer (bulk
soil samples: VARIO MAX CNS Elementar Analysensysteme, Hanau, Germany; silt and clay



fraction: EA3000 CHNS-O Analysis, EuroVector, Milan, Italy). Since the soil contained no
inorganic carbon, carbon contents are equivalent to total organic carbon contents.

**2.3 Biological parameters**

**2.3.1 Root biomass**
All samples were soaked in water and cleaned from soil residues using a sieve of 0.25 mm mesh
size. Fine roots ($\leq$2 mm diameter) longer than 10 mm were extracted manually with tweezers
and subsequently inspected under a stereomicroscope. Living (biomass) and dead fine roots
(necromass) were distinguished by root surface and periderm color, tissue elasticity, cohesion
of cortex, and periderm and stele (e.g. Hertel et al., 2013). The separated fine root biomass and
necromass was dried at 70 °C for 48 h and weighed. While this method displays fine root
biomass with sufficient accuracy, the negligence of root fragments <10 mm length may lead to
an underestimation of fine root necromass. Therefore, the mass of dead fine roots was corrected
for this smaller root fraction by extrapolation using soil depth-specific regression equations that
relate the mass of small dead roots <10 mm length to that of large dead roots >10 mm length.
These regression equations were established for other samples from the same site by analyzing
the mass of small dead roots following a method introduced by van Praag et al. (1988) and
Hertel (1999).
In this study the results of the root biomass and necromass were combined (root mass), since
no significant differences in $^{14}$C contents of root biomass and root necromass were expected
because the forest was established in 1916. Moreover, results of previous studies show that both
living and dead fine root biomass has similar $^{14}$C signature in soil profiles (Gaudinski et al.,
2001; Gaul et al., 2009).

**2.3.2 Microbial biomass carbon**
The microbial biomass carbon ($C_{mic}$) was determined using the chloroform fumigation
extraction (CFE) method (Vance et al., 1987). Briefly, ethanol-free chloroform was used to
fumigate fresh soil of 10 g for 24 h. After removing the chloroform, 40 ml of 0.5 M $K_2SO_4$
solution was added to the soil, which was shaken for 30 min on a horizontal shaker at 250 rev
min$^{-1}$ and centrifuged for 30 min at 4420 x g. A second subsample was treated similarly but
without fumigation. OC concentrations in the supernatants are measured using a TOC-TNb





Analyzer Multi-N/C 2100S (Analytik Jena, Jena, Germany). 200 µl of 1 M HCl was added to
the dilutions to remove inorganic C. Finally $C_{mic}$ was calculated from the difference between
OC of the fumigated and the not fumigated samples using a conversion factor ($k_{EC}$) of 0.45
(Joergensen, 1996). Additionally the ratio of $C_{mic}$ to the total OC content in percentage was
determined to obtain information on the microbial abundance (Agnelli et al., 2004; Anderson
and Domsch, 1989; Bauhus and Khanna, 1999).

### 2.4 Particle size parameters

The density and particle size fractionations were performed with 30 g soil according to Angst
et al. (2016). First, the soil samples were saturated with a sodium polytungstate solution (TC
Tungsten compounds, Germany) with a density of 1.8 g cm$^{-3}$, which was subjected to sonication
(600 J ml$^{-1}$) to break up soil aggregates and release particulate organic matter occluded in soil
aggregates (oPOM). After sonication the POM fraction was removed using a water jet pump.
The remaining mineral residue was repeatedly washed with de-ionized water until the
conductivity of the eluted water was below 50 µs and then wet sieved to obtain the combined
coarse and medium sand (200-2000 µm), fine sand (63-200 µm), and coarse silt (20-63 µm)
fractions. The mineral soil that passed through all three sieves, i.e. the medium silt, fine silt and
clay fraction, was subjected to sedimentation to separate the medium silt (6.3-20 µm) from the
combined fine silt and clay fraction (<6.3 µm). All fractions were freeze-dried for further
analysis. The density and particle size fractionations were performed on samples down to 110
cm depth.

### 2.5 Radiocarbon analysis

Radiocarbon analysis was performed on bulk SOM and on the fine silt and clay fraction (<6.3
µm). Prior to the analysis, visible plant residues were removed from the bulk soil under a
microscope using tweezers. All samples were treated using a modified protocol according to
Rethemeyer et al. (2013). Briefly, all samples were extracted with 0.5 % HCl (instead of 1 %
HCl) first for one hour at 60 °C and then over night at room temperature. HCl was removed by
washing with Milli-Q water. After drying, the samples were combusted and graphitized using
an elemental analyser for sample combustion (Rethemeyer et al., 2013), which limits the
amount of sample that can be weight into the 8 x 8 x 15 mm small tin boats. $^{14}$C contents were
measured on a 6 MV Tandetron AMS (HVE, The Netherlands) at the University of Cologne





(Dewald et al., 2013). The results of the [14]C measurements are reported in percent modern
carbon (0 pMC, related to 1950) with one-sigma uncertainties.

**2.6 Statistical methods**
The correlation between all soil parameters was analysed using a PCA performed with the
software PAST 3.06 for Windows (Hammer et al., 2001). The data set was reduced to 14
samples (profiles A and D: 10-110 cm, profile G: 10-60 cm, and 110 cm) by removing those
with missing values of some soil parameters. The Kruskal-Wallis test was applied to ensure that
all soil profiles (A, D, and G) of the sampling site originated from the same population
(significant when $p < 0.05$). If necessary, the variables were transformed to ensure their normal
distribution, which was tested using the Shapiro-Wilk test (significant when $p < 0.01$). The
statistical tests were performed using R 3.2.0 software (R Core Team, 2015). All measured
parameters were standardized (centered and scaled) to ensure their comparability. Furthermore,
the average and absolute deviation (range) of [14]C values (bulk OC and silt & clay fraction)
including measurement errors were calculated from the three soil profiles. These data reflect
the combined variability at each sampling depth of the three soil profiles (A, D, and G).

**3. Results**

**3.1 Elemental composition of SOM**
In each soil profile of the transect OC contents decrease significantly with increasing soil depth
from maximal values of 1.54 and 1.69 % in the Bsv horizons (10 cm) of the three profiles to
minimal values of 0.02 % (Supplement Tab. S1; Fig. 2). OC contents decrease strongly below
the Bsv horizon to values of 0.49 - 0.71 % at 35 cm depth (Bv horizon). At 60 cm depth (Cv
horizon) the OC contents are even lower ranging between 0.09 to 0.26 %, with highest values
in the profile closest to the beech. In the IICv horizon at 85 to 110 cm soil depth OC contents
are extremely low with values between 0.04 and 0.17 % which show no clear relation to the
distance from the tree. OC contents increase slightly to 0.26 % in the IIICv horizon at 160 cm
depth in profile A close to the beech. This trend is also observable in profile D with 0.13 % OC
at 135 and 160 cm and in profile G at 135 cm depth (0.17 % OC).
The N contents (Supplement Tab. S1) show a comparable depth distribution to the OC contents
with values ranging from 0.002 to 0.072 %. These OC and N distributions result in C/N ratios



in the range of 7 to 28 which decrease with increasing depth in the profiles D and G, whereas
C/N ratios in profile A (closest to the beech) scatter strongly in a range of 10 to 28.

**3.2 Biological parameters**
The biological parameters, which were analysed include root biomass and necromass and
microbial biomass-derived carbon (Supplement Tab. S1, Fig. 2). The root mass density is
highest in the uppermost Bv and Bsv horizons and varies between 0.9 and 2.5 g $l^{-1}$ soil. In the
uppermost horizon (at 10 cm depth) root mass decreases with increasing distance from the
beech stem from 1.2 to 2.5 g $l^{-1}$ soil. At greater depth, no trend related to distance from the
beech could be observed. No roots could be determined in the IICv horizon at 85 to 110 cm
depth, whereas at greater depth (>110 cm depth) root masses of 0.4 to 1.0 g $l^{-1}$ soil are present.
$C_{mic}$ contents are highest in the uppermost Bsv horizon (10 cm). In the Bsv horizon $C_{mic}$
increases by about 49 % with increasing distance to the beech (from 105 to 216 µg $g^{-1}$ dry
weight - DW; Supplement Tab. S1, Fig. 2). $C_{mic}$ contents decrease strongly in the Bv and C
horizon and show a considerable spatial variability in some soil profiles. The strongest
variability is observed in profile D where $C_{mic}$ contents decline from 203 to 14 µg $g^{-1}$ DW in 35
to 60 cm depth, then increase in 85 cm (125 µg $g^{-1}$ DW), and stay constant at relatively low
concentration (19-26 µg $g^{-1}$ DW) in 110 to 160 cm depth. This variability is not obviously
related to other soil parameters investigated. In profile A, closest to the beech, $C_{mic}$
concentration decline gradually (from 105 to 5 µg $g^{-1}$ DW) in 10 to 85 cm depth and increase
slightly in 110 and 135 cm depth before declining again. In profile G, most distant from the
tree, $C_{mic}$ contents decrease strongly in 10 to 35 cm but slightly increase again in 135 cm.
The contribution of the microbial biomass to SOM - as an indicator of SOM quality and
availability (Anderson and Domsch, 1989; Sparling, 1992) - was determined by the $C_{mic}$/OC
ratio (Tab. 1). This ratio ranges between 0.6 and 3.3 % in the two B horizons (5-45 cm), which
is well in the range of values determined in temperate regions for which data are available (0-
30 cm; Serna-Chavez et al., 2013). The $C_{mic}$/OC ratio increases with increasing distance of the
tree only in the Bv horizon while in the IICv and IIICv horizons at 110 and 135 cm depth values
decrease with increasing distance to the beech. In the five Cv horizons from each profile
investigated the $C_{mic}$/OC ratio is more variable (0.2-23.6 %). Very high ratios of 10.1 and 23.6
% were determined in profiles D at 85 cm depth and profile A at 135 cm depth reflecting a high
abundance of microbial biomass relative to soil OC.



### 3.3 Particle size distribution

The grain size distribution, which was analysed using the protocol described in chapter 2.4, was measured down to 110 cm soil depth shows considerable differences in the distribution of the sand and coarse silt fractions in the three profiles. While the medium and the fine silt and clay fraction decrease or stay constant with depth in profiles A, D, and G (Supplement Tab. S1, Fig. 2), the fine sand fraction strongly increases from 209 and 228 g kg$^{-1}$ soil in 10 cm to 654 and 836 g kg$^{-1}$ soil in 110 cm depth. Coarse and medium sand contents show a decreasing trend with depth in all profiles. The coarse silt fraction is more variable in the three profiles in the range of 118 to 224 g kg$^{-1}$ soil with highest contents in the B horizons (10 and 35 cm). Coarse silt contents are lowest in 85 and 110 cm (except in profile A) with 18 to 93 g kg$^{-1}$ soil. Medium silt and fine silt plus clay contents decrease with increasing depth in all profiles. The medium silt contents range from 15 to 122 g kg$^{-1}$ soil and those of the fine silt and clay fraction from 25 to 66 g kg$^{-1}$ soil.

### 3.4 Radiocarbon contents in soil profiles

$^{14}$C contents of bulk OC vary in the range of 32.6 to 105.0 pMC (Supplement Tab. S1, Fig. 2), which is equivalent to apparent $^{14}$C ages of >modern (post 1950, containing bomb-$^{14}$C) to 9000 years BP. Concentrations decrease in all profiles in 10 to 60 cm soil depth (except in 35 cm of profile D) but stay constant or increase at greater depth. Similar to the distribution of the root mass, $^{14}$C contents decrease in profile A in 10 to 135 cm from 101.6 to 46.0 pMC, but increase again at the lowermost sampling depth of 160 cm to 85.5 pMC. In profile D $^{14}$C contents decrease strongest in 10 to 85 cm depth from 100.1 to 32.6 pMC. The large drop in $^{14}$C at 85 cm depth is related to strong increase of coarse and medium sized sand and decrease of coarse silt. In 110 to 160 cm depth, $^{14}$C contents rise again from 60.5 to 66.5 pMC parallel to increasing amounts of root mass. In profile G $^{14}$C contents decrease in 10 to 110 cm depth from 100.9 to 49.6 pMC, increase again at 135 cm (71.8 pMC) before they drop to 48.7 pMC at 160 cm related also to the distribution of the root mass at these depths.

The $^{14}$C contents of the combined fine silt and clay fraction (<6.3 μm) show decreasing values with increasing depth in all soil profiles. In the B horizons (10 to 35 cm) the $^{14}$C concentrations are slightly lower or nearly equal to bulk OC. In the C horizons (below 35 cm) this fine fraction yields higher $^{14}$C concentration than bulk OC which decrease continuously to lowest values of 59.8 to 67.3 pMC at 110 cm, the lowest depth analysed. This indicates a higher contribution of younger SOM to this fraction with increasing depth.



For each sampling depth of profile A, D, and G average $^{14}$C values of bulk OC and of the fine
silt and clay fraction and their absolute deviation (including measurement uncertainties, see 2.6)
were calculated (Tab. 2). The aim of this approach was to derive information on the spatial
variability of $^{14}$C contents in the different sampling intervals, i.e. soil horizons. These average
$^{14}$C contents of bulk OC decrease in 10 to 85 cm from 100.9 to 52.5 pMC and slightly increase
in 110 to 160 cm soil depth from 55.9 to 66.9 pMC. The absolute deviation of these average
$^{14}$C contents increase strongly with increasing depth from ±1.2 (10 cm) to ±20.5 pMC (85 cm),
with one exception in 110 cm (±6.5 pMC). The highest variability of $^{14}$C contents can be
observed in the C horizons.
The average $^{14}$C contents of the fine silt plus clay fraction decrease less pronounced with
increasing depth (from 100.0 to 62.6 pMC) than that of bulk OC. The absolute deviation is
much lower compared to that of bulk OC ranging from ±1.1 (60 cm) to ±5.5 (85 cm) and
showing no trend related to soil depth.

**305 3.5 Principle component analysis (PCA)**

A PCA was performed to evaluate the correlation and therefore the influence of the different
soil parameter on each other. The two principle components (PC) explain in summary 84.2 %
of the data variation (PC 1 = 70.0 % and PC 2 = 14.2 %; Fig. 3). All parameters, with the
exception of the sand fractions, are positively correlated with PC 1. These parameters all
promote high soil OC contents including N content, root mass, $C_{mic}$, and silt and clay size
fractions. Parameters correlate with $^{14}$C of bulk OC include the coarse silt fraction, which shows
a strong positively correlation followed by N content < root mass < OC content < medium silt
fraction. $C_{mic}$ seems to have a smaller effect on the $^{14}$C of bulk OC but is more closely related
to $^{14}$C of the silt and clay fraction. The $^{14}$C content this fraction also correlates positively with
the medium silt fraction < OC content < root mass < N content, and the coarse silt fraction. PC
2 is strongly affected by the sand content. The negative correlation with the coarse and medium
sand fraction and the positive relation to the fine sand fraction suggests that PC 2 represents the
coarse and organic poor mineral soil matrix.

**320 4. Discussion**


**322 4.1 Influence of root-derived OC on $^{14}$C distribution**



The depth distribution of [14]C contents of SOM is assumed to be significantly affected by the
input of plant-derived OC as the dominant carbon source of SOM. While OC contents in surface
soils are largely controlled by the input of aboveground plant litter, subsoils receive OC mainly
from root biomass and to a minor extend from particulate and dissolved OC transported through
the soil profile (Baisden and Parfitt, 2007; Chabbi et al., 2009; Fröberg et al., 2007, 2009; Rasse
et al., 2005). Roots were found to introduce relatively fresh OC into deep soil horizons with
>modern [14]C contents equivalent to <20 years (Gaudinski et al., 2001; Gaul et al., 2009;
Trumbore et al., 2006). This can cause rejuvenation effects of SOM, i.e. lead to younger
apparent [14]C ages even at greater soil depth, which is particularly important near root channels
(Bundt et al., 2001; Chabbi et al., 2009).
The importance of roots as an OC source to SOM is confirmed by the strong positive correlation
of OC contents with the distribution of the root mass in the three soil profiles (Fig. 2 and 3).
However, the highest root mass was determined in the uppermost subsoil horizons, the Bsv (5-
15 cm) and Bv horizon (15-45 cm). In the C horizons below (45-160 cm), the root mass per soil
volume declines strongly by about 40 to 100 %. The low OC input from living and dead roots
is reflected by low OC contents, which decrease strongly from 1.5-1.7 % (in 10 cm) to minimum
values of 0.02 % in the deeper subsoil. No roots could be detected between 85 and 110 cm
depth probably due to textural changes in the IICv horizon and resulting shortage in plant-
available water (Schenk and Jackson, 2005). Here, a shift toward coarser grain size occurs with
increasing amounts of coarse and medium sand and decreasing amounts of coarse and medium
silt which reduces the storage capacity for plant-extractable water. The recurrence of live and
dead roots in the IIICv and IVCv horizon at 135 and 160 cm depth, respectively, may be
associated again with a change in soil texture, i.e. higher silt contents.
The strong influence of the distribution of roots on [14]C contents is supported by the PCA
analysis revealing a close correlation of both parameters (Fig. 3). Both, the root mass and the
[14]C content of SOM decrease significantly below 35 cm depth. Selectable roots were found
again in the IIICv and IVCv horizons and these are most probably responsible for the increase
in the apparent [14]C ages of bulk SOM, i.e. the rejuvenation of SOM in these lowest horizons
investigated (Fig. 2). [14]C concentrations however, show a considerable variability in 60 to 135
cm depth were only few or no roots could be separated indicating that other soil parameters of
of importance in these carbon-poor subsoil horizons.
In summary, the [14]C concentrations of SOM in the three profiles cannot exclusively be
explained by the distribution of roots. [14]C contents of bulk SOM shows a quite larger variability
compared to that of OC contents and of the root mass suggesting that other factors may be of



importance like soil texture (not investigated below 110 cm depth), mineralogical changes and
associated stabilization effects, and the microbial activity which could influence $^{14}$C contents
of SOM.

**4.2 Effect of microbial biomass distribution on $^{14}$C of SOM**
Previous studies indicate that the activity, abundance, and diversity of the microbial biomass
decreases significantly in subsoil horizons most probably due to the reduced OC content and a
lower substrate quality at greater soil depth (Agnelli et al., 2004; Fierer et al., 2003; Fontaine
et al., 2007; Struecker and Joergensen, 2015; Taylor et al., 2002). This may promote high
apparent $^{14}$C ages of bulk SOM. Similar to the results of these studies, we also found strongly
declining $C_{mic}$ contents below 35 cm (below the B horizons; Supplement Tab. S1), which most
probably indicate less favourable conditions for microorganisms at greater depth. However, in
some profiles $C_{mic}$ increases at greater soil depth including profile A (110 cm: 36.6 µg g$^{-1}$ DW,
135 cm: 51.8 µg g$^{-1}$ DW) and profile D (85 cm: 125.3 µg g$^{-1}$ DW; Supplement Tab. S1). These
relatively high $C_{mic}$ values result in high $C_{mic}$/OC ratios (Tab. 1), which suggest that here the
organic matter is more bioavailable than in other layers of the Cv horizons. The very high C/N
ratio of 19 and 29 in profile A (110 and 135 cm) at these potential hot spots, however, does not
support a high bioavailability of the SOM suggested by the $C_{mic}$/OC ratio. Moreover, no roots
were present here which may represent a source of fresher, microbe- available OC which is
necessary for establishing hot spots (Kuzyakov, 2010; Bundt et al., 2001; Sanaullah et al.,
2011). Thus easily degradable OC may have been introduced into the Cv horizon as DOC
through preferential flow pathways. Increasing $^{14}$C contents in 135 cm (profile D and G) and
160 cm depth (profile A) suggest the presence of fresh substrate which most probably is due to
a higher abundance of roots (see 4.1), but the higher root mass does not result in higher $C_{mic}$
contents.
These observations and the results of the PCA analysis reveal that microbial-derived carbon
does not promote higher $^{14}$C contents of bulk OC (Fig. 3). However, $C_{mic}$ values correlate much
stronger with the $^{14}$C contents of the organic-rich fine silt plus clay fraction. This suggests that
microbial-derived OC is potentially stabilised by interaction with fine silt and clay particles.
Comparable results were obtained by Rumpel et al. (2010) for a Podzol and a Cambisol under
forest. Here, microbial-derived polysaccharides were enriched in the mineral fraction (>2 g cm$^{-}$
$^{3}$). However, the supposedly microbial-derived, mineral-bound OC in the study of Rumpel et





al. (2010) had similar to slightly lower [14]C concentrations than the bulk SOM indicating that
OC of microbial origin is stabilized over longer time scales by organo-mineral interaction.
In the soil profiles of this study, the fine silt and clay fraction yields higher [14]C contents than
bulk OC in 60 to 110 cm soil depth (where the root mass is lowest) suggesting a relatively fast
turnover of the younger, potentially microbial OC. The difference in the [14]C contents of bulk
OC (32.6 pMC) and the fine silt and clay fraction (66.7 pMC) is most pronounced in profile D
at 85 cm  (Supplement Tab. S1) and probably indicates a region with limited access to fresh
(young) SOM. However, the weak correlation of $C_{mic}$ with the other soil parameters analysed,
including the root mass and OC content (Fig. 2, Fig. 3) suggests a minor influence of $C_{mic}$ on
the [14]C concentrations of SOM.

**4.3 Influence of grain size on [14]C contents**
Soil texture, particularly small particle sizes (<2 μm), may considerably influence the [14]C
concentration of SOM. Their large surface area with which the organic matter can form organo-
mineral assemblages promote the protection of OC against microbial and oxidative degradation
(e.g. Kleber et al., 2005; Kögel-Knabner et al., 2008; Mikutta et al., 2006; Spielvogel et al.,
2008) thus resulting in high OC contents and low [14]C concentrations (von Lützow et al., 2006;
Rumpel et al., 2004; Trumbore, 2009). This relationship is reflected by a strong positive
correlation of the fine silt and clay fraction (<6.3 µm) with the [14]C content of bulk OC (Fig. 3).
However, the decline of the silt and clay fractions with depth is less variable than the depth
distribution of the [14]C concentration of bulk SOM.
If interaction of OC with soil minerals is an important stabilization mechanism in subsoil, then
the [14]C concentration of the fine silt and clay fraction should be lower compared to that of bulk
OC. This fraction however, has similar or only slightly lower [14]C contents in the B horizons (10
and 35 cm) and higher contents in the C horizons (60-110 cm) compared to those of bulk OC.
These results indicate that younger OC sources, most likely microbial-derived OC are
associated with fine silt and clay particles in the C horizons (see 4.3). Moreover, the association
of OC with small grain-sizes may be less strong than assumed in previous studies resulting in
higher turnover times and [14]C contents, respectively. Likewise, relatively high [14]C contents
were determined for a soil fraction extracted with hydrofluoric acid (HF) which was thought to
be the most strongly mineral associated and thus the oldest SOM fraction (Eusterhues et al.,
2007). The authors of this study suggested that stabilization by interaction with the mineral
matrix is less effective than other stabilization mechanisms like recalcitrance of organic



compounds and occlusion of SOM in soil aggregates. They further assumed that the mineral-
associated SOM may be diluted by fresher (younger) SOM or the HF-resistant SOM may reflect
stable OC, respectively. Therefore, the higher [14]C contents of the fine silt and clay fraction can
also be a result of the sorption of younger SOM at mineral surfaces or the exchange of older by
younger SOM.
The major particle size change in the analysed profiles is that of the sand fractions, which
however, stores only little OC and thus does not influence the [14]C distribution (Angst et al.,
2016; von Lützow et al., 2006; Trumbore, 2009). This is confirmed by the PCA showing that
the sand fractions does not correlate with any other soil parameter influencing the SOM
distribution, including OC content, root mass, $C_{mic}$, and the particle size fractions <63 µm
(Supplement Tab. S1, Fig. 2, and 3).
In summary the [14]C distribution of bulk OC in the soil profiles is mainly affected by the silt and
clay sized fractions because these fractions contain the majority of the SOM (Angst et al., 2016).

**4.4 Spatial heterogeneity of [14]C contents**
The distribution of the organic matter in the subsoil is thought to be spatially very
heterogeneous due to preferential flow paths and local root branches through which OC is
transferred into deeper soil horizons (Bundt et al., 2001; Chabbi et al., 2009; Don et al., 2007;
Salomé et al., 2010; Syswerda et al., 2011). Accordingly, [14]C concentrations of subsoil organic
matter are expected to be even more heterogeneously than those in surface soils. This was
shown in a study of Chabbi et al. (2009) who found close to modern apparent [14]C age near
preferential flow paths while the OC of the surrounding soil was several thousand years old.
The variability of [14]C concentrations of bulk OC in the three soil profiles analysed confirm the
supposed large spatial heterogeneity in subsoil horizons even on the small scale of a three meter
long soil transect (Fig. 2, Tab. 2). The absolute deviation of [14]C contents of bulk OC calculated
for each horizon of the three profiles (Tab. 2) indicates a much larger variability in the C
horizons compared to the B horizons (except in IICv at 110 cm). In contrast the fine silt and
clay fraction show a much smaller variability in their [14]C contents in each horizons (Tab. 2)
most probably because larger roots have been removed from this fraction by sieving and density
separation. The large spatial [14]C variability in the deeper C horizons analysed (135 and 160 cm;
Tab. 2) thus may be caused by the strong effect of younger root-derived OC on the relatively
low [14]C concentrations of bulk SOM.



The microbial biomass may influence the spatial $^{14}$C distribution of bulk SOM indirectly by the
mineralization of fresher (younger) OC resulting in low $^{14}$C concentration (Supplement Tab.
S1, similar to priming effects, e.g. Kuzyakov, 2010). This can lead to a relative enrichment of
older organic compounds.


**5. Conclusion**
In this study, the influence of roots (bio- and necromass) and its OC input was identified as
major factor affecting the spatial $^{14}$C distribution of SOM in subsoil horizons of a Dystric
Cambisol under beech forest. The distance of the three soil profiles analysed to a beech did not
affect the spatial distribution of roots and of $^{14}$C contents. Other soil parameters including soil
texture and the microbial biomass had no statistically significant influence on the $^{14}$C
distribution of bulk SOM. Organic matter included in silt and clay sized particles (<6.3 μm),
which were thought to stabilise OC on longer time scales, had slightly higher $^{14}$C contents in
the C horizons than bulk OC and may contain younger, microbial-derived compounds. Thus, in
contrast to previous studies, OC stabilization by organo-mineral interaction seems to be of
minor importance in this sandy subsoil. We did not observe a continuous increase of apparent
$^{14}$C ages with depth as in most previous studies, but a large horizontal as well as vertical $^{14}$C
variability in the three soil profiles. High apparent $^{14}$C ages of up to 9000 yrs BP may be a result
of a reduced microbial activity or the lack of easily degradable SOM at greater depth. $^{14}$C
contents of bulk SOM are most variable in the C horizons because of large differences in the
abundance of root mass. The fine silt and clay fraction (<6.3 μm) yields less heterogeneous $^{14}$C
contents due to the absence of larger root fragments and thus may be a more reliable indicator
of humified SOM which is less strongly influenced by fresh carbon inputs. These results
indicate that estimates of soil OC turnover based on $^{14}$C analysis of bulk SOM in an individual
soil profile may be misleading as they mainly reflect the local distribution of roots.

**Acknowledgements**
This project was funded by the German Science Foundation (DFG) within the Research Unit
1806 "The forgotten part of carbon cycling: Organic matter storage and turnover in subsoil".
We thank Ulrike Patt, Elisabeth Krewer, and Anna Distelrath for their help with $^{14}$C sample
processing. Thanks also to Stefan Heinze and Alfred Dewald for high quality AMS





measurements and all colleagues of the DFG Research Unit for their cooperation and helpful
discussions.

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



**Table 1**: Ratio of microbial biomass-derived carbon ($C_{mic}$) to total OC
content in profiles A, D, and G.

| soil depth (cm) | horizon | $C_{mic}$ / OC (%) | | |
|---|---|---|---|---|
| | | **A** | **D** | **G** |
| 10 | Bsv | 0.6 | 1.1 | 1.4 |
| 35 | Bv | 0.9 | 3.3 | 0.4 |
| 60 | Cv | 0.6 | 1.1 | 0.9 |
| 85 | II Cv | 0.8 | 23.6 | - |
| 110 | II Cv | 7.9 | 6.4 | 0.2 |
| 135 | III Cv | 10.1 | 1.4 | 0.7 |
| 160 | IV Cv | 0.2 | 2.0 | - |




**Table 2**: Average $^{14}$C values of bulk OC and of the silt and clay
fraction (<6.3 μm) and absolute deviations for the three soil
profiles A, D, and G (n = 3).

| depth (cm) | horizon | $^{14}$C bulk OC (pMC) | $^{14}$C fine silt & clay (pMC) |
|---|---|---|---|
| 10 | Bsv | 100.9 ± 1.2 | 100.0 ± 1.5 |
| 35 | Bv | 97.8 ± 7.9 | 93.1 ± 2.5 |
| 60 | Cv | 65.2 ± 10.1 | 81.3 ± 1.1 |
| 85 | II Cv | 52.5 ± 20.5 | 71.6 ± 5.5 |
| 110 | II Cv | 55.5 ± 6.5 | 62.6 ± 3.1 |
| 135 | III Cv | 60.3 ± 15.1 | - |
| 160 | IV Cv | 66.9 ± 18.9 | - |




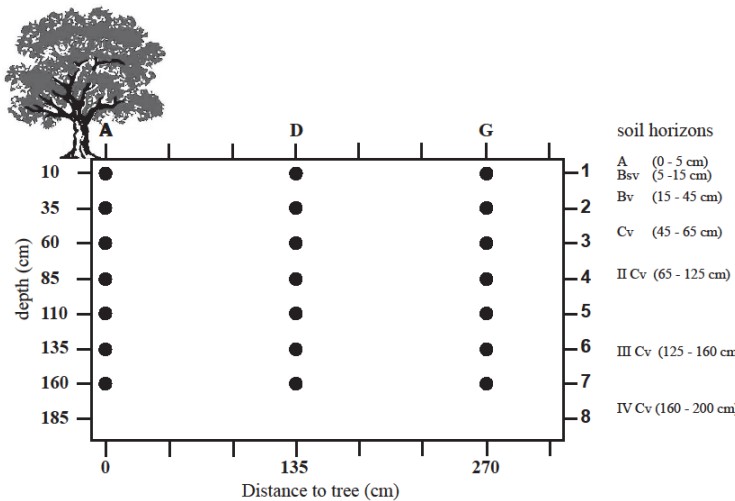


**Figure 1**: Sampling design of the soil transect. Analysed samples from profiles A (0 cm distance to the beech), D (135 cm distance), and
G (270 cm distance) are displayed by black dots.





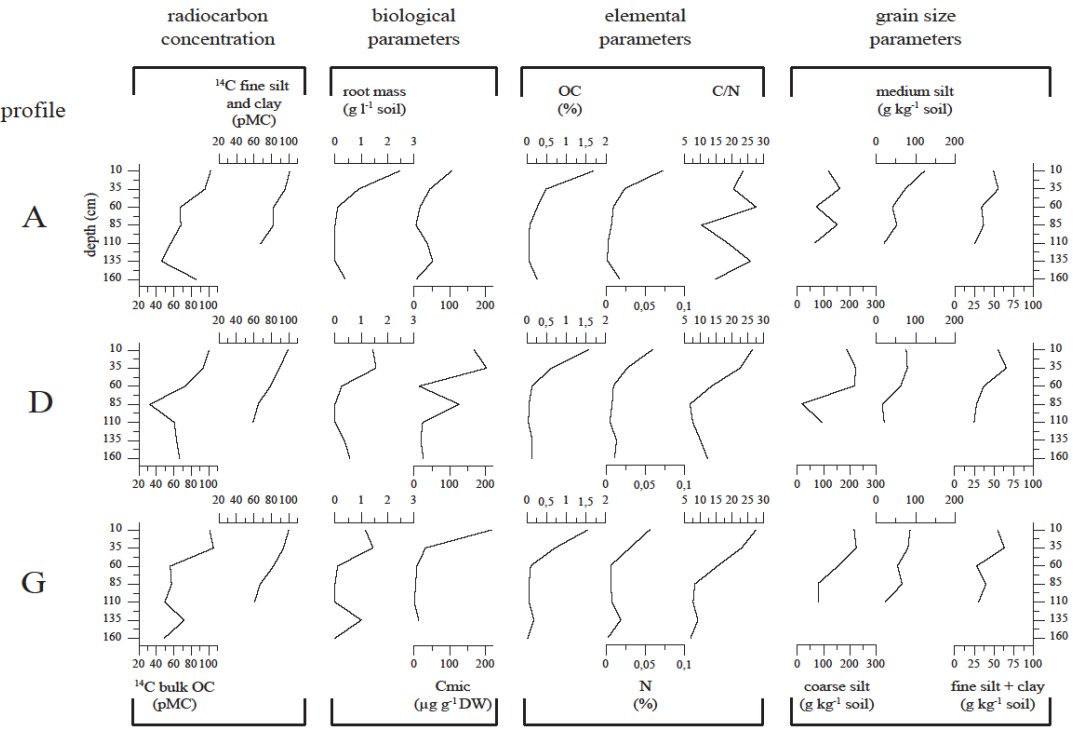


**Figure 2**: Selected soil parameters affecting $^{14}$C concentrations of bulk SOM in the three profiles A, D, and G (see Supplement Tab. S1).





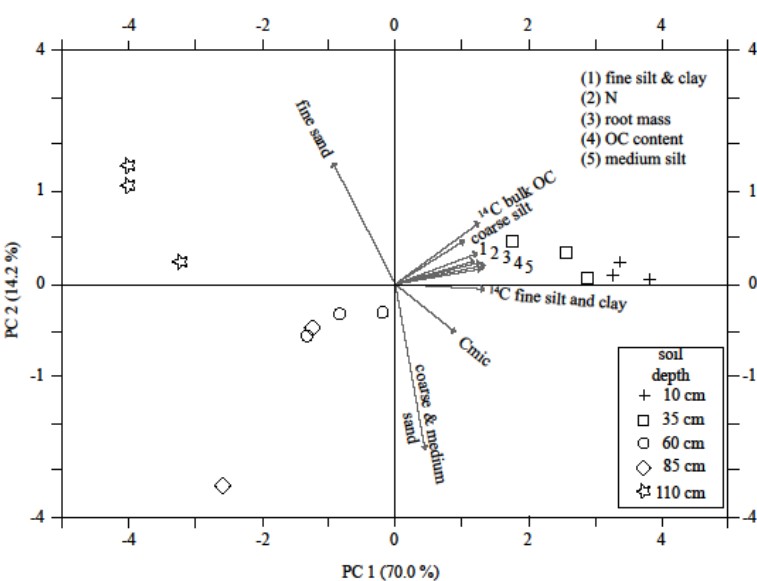


**Figure 3**: PCA biplot of measured soil parameters in samples from different soil depth (represented by symbols). Some parameters are represented

by numbers explained in the legend above.