# Peer review of "Which are important soil parameters influencing the spatial heterogeneity of"

_Biogeosciences, 2016_

## Referee Comment (RC1) · Anonymous Referee #1 · 8 Mar 2016

Review of MS Which are important soil parameters influencing the spatial heterogeneity of 14C in soil organic matter? by John et al.

The authors analyzed the 14C content of organic carbon in bulk soil and soil fractions at different distances from a beech tree. They discuss possible driving factors for the measured variability. This is an interesting and relevant topic, and the presented data deserve publication. However,

1) The study is rather descriptive and misses a sound statistical analysis. The approach makes it difficult to identify driving factors for 14C variability because many explanatory variables are cross-related. For example, soil depth, pMC, root biomass and silt/clay content correlate significantly with each other, hence, assignment of unique factors to pMC variability is hampered. Authors should consider using a general linear model for

their analysis.

2) It is not clear why authors explicitly excluded A-horizons from their study.

3) Without any estimate of C input from aboveground and belowground litter, i.e., without adding a dynamic component to the study, the explanation of 14C distributions remains vague.

Further, grammar and syntax are partially poor and I recommend copy-editing of the text.

Detailed comments

Title: The title should reflect that only one single gradient at different distances from a beech tree was studied. Hence, at least the word 'forest' should appear in the title.

Line 38. Why does sentence begin with 'however'?

Line 58. The description of parameters influencing the soil's 14C content is not well structured. The 14SOC content is in principal determined by i) input rates, ii) turnover or loss rates, iii) radioactive decay, and iv) changes in atmospheric $14CO_2$, i.e., changes in 14C of the input material over time. Soil or vegetation or climate properties modulate some of these principal factors. Authors are requested to put their list of factors into a logical order following these drivers.

Line 64. Is heterogeneity of SOM age meant?

Line 65 and later. Input from roots is called root litter, from living roots rhizodeposition. This is not in contrast to being 'fresh'.

Line 72 .There are studies on spatial variability of 14C in soils, e.g. Leifeld and Mayer (2015), Budge et al. (2011); Schoning et al. (2013). Results in these publications revealed also different patterns as compared to the current study.

Line 75-82. Again, the argumentation is not stringent. All listed factors just reflect a

change in SOM turnover rates, albeit caused by different mechanisms.

Line 84 and elsewhere. Authors are requested to explain what 'apparent 14C age' and, later 'apparent MRT' refers to.

Line 99. I suggest adding 'of a forest soil' behind 'subsoils'.

Line 183 ff. It is not clear why soils were treated with 0.5 % HCl before AMS. This removes part of SOM of a particular but unknown signature. If amount and signature of the dissolved C is not the same for all samples, the resulting 14Csample may lead to biased conclusions.

Line 336. Strictly spoken, the root biomass does not tell very much about the input from roots because root turnover and rhizodeposition may change with depth. Also the correlation between % SOC and root biomass must not be conclusive. Higher SOC in topsoils is, at least partially, resulting from aboveground litter input, and this explains much of the typically found difference in the mass-depth slope between depth and % SOC on the one hand and depth and root biomass on the other (see e.g. Jackson et al. (1996); Jobbagy and Jackson (2000).

Line 354. This conclusion is difficult to draw without consideration of carbon input rates. Chapter 4.2. I think the attempt to explain 14C by microbial biomass parameters is highly misleading. The measured microbial biomass reflects the current situation and its turnover time is in the range of months whereas 14SOC integrates processes that took place over centuries and millennia. The authors implicitly assume that the Cmik distribution in their profiles is representative for much longer timescales, which they do not know.

Line 366. Authors may also consider that DOC ages during its journey through the soil column; this may increase its 14C age substantially.

Line 385. Sentence unclear.

Line 437-444. I would argue that, in addition, the smaller topsoil variability in 14C

reflects the important role of aboveground litter inputs, which may be similar among the three sites.

Line 456. This is in some contradiction to line 383.

Line 461. OC input has not been estimated in this study, which is a major shortcoming. Hence, authors should not refer to input as a driving force for 14SOC unless they do a proper input estimate.

References

Budge, K., Leifeld, J., Hiltbrunner, E., and Fuhrer, J. (2011). Alpine grassland soils contain large proportion of labile carbon but indicate long turnover times. Biogeosciences 8, 1911-1923.

Jackson, R. B., Canadell, J., Ehleringer, J. R., Mooney, H. A., Sala, O. E., and Schulze, E. D. (1996). A Global Analysis of Root Distributions for Terrestrial Biomes. Oecologia 108, 389-411.

Jobbagy, E. G., and Jackson, R. B. (2000). The Vertical Distribution of Soil Organic Carbon and Its Relation to Climate and Vegetation. Ecological Applications 10, 423-436.

Leifeld, J., and Mayer, J. (2015). 14C in cropland soil of a long-term field trial – experimental variability and implications for estimating carbon turnover. SOIL 1, 537-542.

Schoning, I., Gruneberg, E., Sierra, C. A., Hessenmoller, D., Schrumpf, M., Weisser, W. W., and Schulze, E. D. (2013). Causes of variation in mineral soil C content and turnover in differently managed beech dominated forests. Plant and Soil 370, 625-639.

---

## Referee Comment (RC2) · Anonymous Referee #2 · 15 Mar 2016

The authors studied the spatial distribution of 14C with depth as well as with distance to a beech tree, and discussed possible factors influencing this distribution, including biological, physical and chemical soil properties.

Although this is an interesting study, I suggest major revision for the following reasons.

- I was wondering why the authors did not include the topsoil. This is surprising as it was said at several passages in the Introduction and Discussion chapters that for topsoil, aboveground litter plays an important role for SOM. Similarly, a negative point is that several parameters were obtained solely down to a depth of 110 cm.

- PCA as the only statistical tool is not sufficient for a comprehensive and sound interpretation, especially when taking into account that some of the studied soil properties may correlate with each other. Here, PCA did not yield unambiguous results, so it is

not safe to declare roots as the main factor, especially as dissolved organic carbon was not investigated.

- It is nowhere explained, why it is desirable to improve understanding of SOM cycling and turnover. One or two sentences would be enough to show the reasons for such research like in the current study.

Detailed remarks:

TITLE

- I miss the information, that the soil under a tree, thus likely forest soil was investigated. This information does not appear before the middle of the Abstract, and towards the end of the Introduction.

ABSTRACT

- Lines 28f: This is a rather sudden transition from general points to the current study.

- Lines 32f: It was not mentioned here that only fine roots were analysed.

- Line 45: "... to determine SOM" sounds strange. Please rephrase.

INTRODUCTION

- I missed some references like Marschner et al. (2008) when talking about different turnover times of SOM compounds (lines 54ff), or Kautz et al. (2013). Although the latter review deals with nutrient uptake from arable and not forest subsoils, it could be included in the current study, as the Introduction is partially written very general.

MATERIAL AND METHODS

- Lines 120ff: If profile A was sampled directly beneath the tree, I am sure that there were also coarse roots present.

- Lines 129f: Please give the temperature used for combustion of the samples.

[Figure]

- Lines 161f: The authors treated the sampes with acid to remove inorganic carbon. This contradicts the statement made in lines 129f.

- Lines 180f: Why were some analyses like particle size not measured on the complete sample set?

RESULTS

- Lines 273ff: In my opinion, it is not necessary to give decimals for the pMC values.

- Lines 312ff: The phrasing with the "<" signs is not easy to read.

DISCUSSION

- Chapter 4.1 has the heading "Influence of root-derived OC on 14C distribution". However, first it begins with repetition from the Introduction as well as citing references rather than discussing the own data, which I find inappropriate. Second, the complete second paragraph discusses the root depth distribution rather than its influence. The actual statement made about root influence on 14C distribution is restricted to the end of the chapter and is written rather vague.

- At the end of chapter 4.3 (lines 433f.), it sounds like grain size would have a stronger effect on 14C distribution than roots, which contradicts the overall statement of the study. Please rephrase.

Further notes:

- Language is partially bad and has to be considerably improved, also to avoid missunderstandings in case of ambiguous statements (e.g. lines 140ff, lines 190ff.).

- Throughout the manuscript, SI units should be used, also for C contents.

- From Fig. 1, I can not clearly see if the horizon boundaries were even or undulating, and if they occurred at identical depth throughout the transect. Maybe a picture would be good.

- I also noticed that the horizon terms refer to the German soil classification system, whereas the soil classification itself refers to WRB. Please use a uniform system.

- The supplement shows the same data like Figure 2. One of the two might be skipped to avoid repetition of the same data.

LITERATURE

Marschner B., Brodowski S., Dreves A., Gleixner G., Gude A., Grootes P.M., Hamer U., Heim A., Jandl G., Ji R., Kaiser K., Kalbitz K., Kramer C., Leinweber P., Rethemeyer J., Schäffer A., Schmidt M.W.I., Schwark L., Wiesenberg G.L.B. (2008) How relevant is recalcitrance for the stabilization of organic matter in soils? J. Plant Nutr. Soil Sci. 171, 91–110.

Kautz T., Amelung W., Ewert F., Gaiser T., Horn R., Jahn R., Javaux M., Kemna A., Kuzyakov Y., Munch J.-C., Pätzold S., Peth S., Scherer H. W., Schloter M., Schneider H., Vanderborght J., Vetterlein D., Walter A., Wiesenberg G. L. B., Köpke U. (2013) Nutrient acquisition from arable subsoils in temperate climates: A review. Soil Biol. Biochem. 57, 1003–1022.

---

## Author Comment (AC1) · 20 May 2016

First of all, we would like to express our thanks for the review. We considered all comments carefully, which significantly improved the general quality and structure of the MS. Below, we will provide a point-to-point reply to the comments.

Sincerely, Stephan John

Comments on anonymous referee (RC#1); published on 8 March 2016

1) The study is rather descriptive and misses a sound statistical analysis. The approach makes it difficult to identify driving factors for 14C variability because many explanatory variables are cross-related. For example, soil depth, pMC, root biomass and silt/clay content correlate significantly with each other, hence, assignment of unique factors to

pMC variability is hampered. Authors should consider using a general linear model for their analysis.

We agree that driving factors for 14C are difficult to identify only by PCA and added linear regression models with 14C as dependent variable. However, the PCA should be maintained, because it is a multivariate statistical approach that makes it possible to identify the interaction of highly correlated soil parameters (Reimann et al., 2008) and contain important information in such a complex (sub-) soil system.

2) It is not clear why authors explicitly excluded A-horizons from their study.

This study was performed in the framework of a larger research project on subsoil carbon dynamics which uses a comparable sampling scheme (starting in 10 cm) at different sites excluding the A horizons (5 cm). We consider the A horizon of being of minor importance since this study should solely focus on subsoil.

3) Without any estimate of C input from aboveground and belowground litter, i.e., without adding a dynamic component to the study, the explanation of 14C distributions remains vague.

Unfortunately, we were not able to estimate C inputs from above- and belowground litter. In contrast to cultivated sites with known C inputs, the determination of C input under natural forest is a parameter, which is extremely difficult to determine. Estimations of C inputs and dynamics in subsoil under forest were widely unknown (Rumpel et al., 2012). Furthermore, natural 14C contents do not only reflect short term dynamics but integrate over longer time scales. Thus, the input of the surface soil is not essential in this study.

4) Further, grammar and syntax are partially poor and I recommend copy-editing of the text.

We improved the grammar and syntax.

Detailed comments

5) Title: The title should reflect that only one single gradient at different distances from a beech tree was studied. Hence, at least the word 'forest' should appear in the title.

We changed the title accordingly.

6) Line 38. Why does sentence begin with 'however'?

The sentence has been changed.

7) Line 58. The description of parameters influencing the soil's 14C content is not well structured. The 14SOC content is in principal determined by i) input rates, ii) turnover or loss rates, iii) radioactive decay, and iv) changes in atmospheric 14CO2, i.e., changes in 14C of the input material over time. Soil or vegetation or climate properties modulate some of these principal factors. Authors are requested to put their list of factors into a logical order following these drivers.

The input of C is the most important factor determining the 14C content of bulk OC. This general statement should be maintained (it includes potential changes over time). We added a remark on the radioactive decay and changes of the atmospheric 14C content. We modified the following sentence listing soil properties and environmental factors, which 'modulate' 14C contents.

8) Line 64. Is heterogeneity of SOM age meant?

We agree that the sentence was written in a rather unclear way. Therefore, we modified the sentence.

9) Line 65 and later. Input from roots is called root litter, from living roots rhizodeposition. This is not in contrast to being 'fresh'.

We agree with the reviewer and use the suggested terms.

10) Line 72 .There are studies on spatial variability of 14C in soils, e.g. Leifeld and Mayer (2015), Budge et al. (2011); Schoning et al. (2013). Results in these publications revealed also different patterns as compared to the current study.

We are thankful for the suggested literature and added some of the suggested manuscripts.

11) Line 75-82. Again, the argumentation is not stringent. All listed factors just reflect a change in SOM turnover rates, albeit caused by different mechanisms.

Unfortunately, this remark is not clear. We listed factor that change SOM dynamics and don't want to go into more detail by naming parameter that modulate these factors. We consider this to be not necessary for a paper, which investigates factors affecting the 14C distribution of SOM.

12) Line 84 and elsewhere. Authors are requested to explain what 'apparent 14C age' and, later 'apparent MRT' refers to.

Apparent means that SOM was defined by Trumbore (2009). It is the 'apparent' age or MRT of the complex mixture of various compounds turning over on different time scales. This explanation can be found in the introduction.

13) Line 99. I suggest adding 'of a forest soil' behind 'subsoils'.

Changed according to your comment.

14) Line 183 ff. It is not clear why soils were treated with 0.5 % HCl before AMS. This removes part of SOM of a particular but unknown signature. If amount and signature of the dissolved C is not the same for all samples, the resulting 14Csample may lead to biased conclusions.

In radiocarbon dating this is a standard procedure to remove any inorganic C from soil samples that may bias its 14C content. This acid treatment does not remove significant amounts of OC as most fulvic acids are removed after alkali treatment which was not applied (see Bräuer et al., 2013). Moreover, we used a more diluted HCl as normally done (0.5 instead of 1%) as suggested by the Groningen AMS lab (presented at the 14C Conference in 2010). Thus 14C contents are not biased because of this treatment.

15) Line 336. Strictly spoken, the root biomass does not tell very much about the input from roots because root turnover and rhizodeposition may change with depth. Also the correlation between % SOC and root biomass must not be conclusive. Higher SOC in topsoils is, at least partially, resulting from aboveground litter input, and this explains much of the typically found difference in the mass-depth slope between depth and % SOC on the one hand and depth and root biomass on the other (see e.g. Jackson et al. (1996); Jobbagy and Jackson (2000).

We agree that the upper 35 cm are still significantly influenced by aboveground OC and this was already stated in the text and is reflected by unpublished lipid biomarker analyses. However, in the subsoil horizons below soil OC contents and 14C contents are strongly affected by the root distribution. Unfortunately, we don't have any data on rhizodeposits and root turnover but our data give enough evidence for our statements given in this chapter. Moreover they confirm results of a previous study by Rasse et al., (2005). We improved the text by making some changes (chapter 4.1).

16) Line 354. This conclusion is difficult to draw without consideration of carbon input rates. Chapter 4.2. I think the attempt to explain 14C by microbial biomass parameters is highly misleading. The measured microbial biomass reflects the current situation and its turnover time is in the range of months whereas 14SOC integrates processes that took place over centuries and millennia. The authors implicitly assume that the Cmik distribution in their profiles is representative for much longer timescales, which they do not know.

Line 354: We listed several parameters that may - in addition to the root mass - affect the 14C distribution. This is no final conclusion but just a speculation which could not be improved by C input data because more detailed studies are required to verify our assumptions.

Chapter 4.2: We agree that Cmic represent the present situation (i.e. viable microbes) and thus is difficult to be compared with 14C data. However, Cmic has already been

used to identify ,hot spots, of microbial biomass Bundt et al. (2001), which could prob-ably have strong effects on 14C, e.g. due to utilization of labile (young) OC. However, there is no other parameter that can easily be measured representing the microbial biomass over longer timescale (lipid biomarkers are also difficult because of selective degradation). Thus we keep on using Cmic as an indicator for the current situation assuming no significant changes in the mature beech forest. We added a comment to this in the text and replace the word 'effect' to 'interaction' in caption 4.2.

17) Line 366. Authors may also consider that DOC ages during its journey through the soil column; this may increase its 14C age substantially.

Unfortunately, DOC fluxes were not included in this study. DOC fluxes were determined in another project in an adjacent study site. They were found to be extremely low and their 14C ages were modern at all depth intervals (unpublished data). This result is comparable to previous studies which determined low DOC fluxes in sandy soils under forest (e.g. Dosskey and Bertsch, 1997; Fröberg et al., 2007; McDowell and Likens, 1988). Thus we assume that DOC has a minor effect on the 14C contents of bulk soil OC.

18) Line 385. Sentence unclear.

We modified this sentence. We compare our result, suggesting that microbial derived OC is enriched in fine particle size fractions, with a study by Rumpel who found - in contrast to our result - that 'microbial-derived polysaccharides were enriched in the mineral fraction (>2 g cm-3).

19) Line 437-444. I would argue that, in addition, the smaller topsoil variability reflects the important role of aboveground litter inputs, which may be similar among the three sites.

We added this argument in chapter 4.4

20) Line 456. This is in some contradiction to line 383.

We rewrote this part to clarify that abundances or content of microbial derived C do not promote 14C contents but that higher abundances of microorganisms can utilize fresh (young) OC, resulting in lower 14C contents.

21) Line 461. OC input has not been estimated in this study, which is a major shortcoming. Hence, authors should not refer to input as a driving force for 14SOC unless they do a proper input estimate.

According to comment 3, we were not able to estimate C input rates. Therefore, we decided to replace the word 'source' inputs by 'source' derived OC.

References

Bräuer, T., Grootes, P. M. and Nadeau, M.-J.: Origin of subsoil carbon in a chinese paddy soil chronosequence, Radiocarbon, 55(2), 1058–1070, doi:10.2458/azu_js_rc.55.16367, 2013.

Bundt, M., Widmer, F., Pesaro, M., Zeyer, J. and Blaser, P.: Preferential flow paths: Biological "hot spots" in soils, Soil Biology and Biochemistry, 33(6), 729–738, doi:10.1016/S0038-0717(00)00218-2, 2001.

Dosskey, M. G. and Bertsch, P. M.: Transport of dissolved organic Matter through a Sandy Forest Soil, Soil Science Society of America Journal, 61(3), 920, doi:10.2136/sssaj1997.03615995006100030030x, 1997.

Fröberg, M., Jardine, P. M., Hanson, P. J., Swanston, C. W., Todd, D. E., Tarver, J. R. and Garten, C. T.: Low dissolved organic carbon input from fresh litter to deep mineral soils, Soil Science Society of America Journal, 71(2), 347, doi:10.2136/sssaj2006.0188, 2007.

McDowell, W. H. and Likens, G. E.: Origin, composition, and flux of dissolved organic carbon in the Hubbard Brook Valley, Ecological Monographs, 58(3), 177, doi:10.2307/2937024, 1988.

Rasse, D. P., Rumpel, C. and Dignac, M. F.: Is soil carbon mostly root carbon? Mechanisms for a specific stabilisation, Plant and Soil, 269(1-2), 341–356, doi:10.1007/s11104-004-0907-y, 2005.

Reimann, C., Filzmoser, P., Garret, R. and Dutter, R.: Statistical data analysis explained : Applied environmental statistics with R, 1st ed., Wiley, J, New York., 2008.

Rumpel, C., Chabbi, A. and Marschner, B.: Chapter 20: Carbon storage and sequestration in subsoil horizons: Knowledge, gaps and potentials, in Recarbonization of the Biosphere, p. 559., 2012.

Trumbore, S.: Radiocarbon and soil carbon dynamics, Annual Review of Earth and Planetary Sciences, 37(1), 47–66, doi:10.1146/annurev.earth.36.031207.124300, 2009.

---

## Author Comment (AC2) · 20 May 2016

First of all, we would like to express our thanks for the review. We considered all comments carefully, which significantly improved the general quality and structure of the MS. Below, we will provide a point-to-point reply to the comments.

Sincerely, Stephan John

Author comments on anonymous referee (RC#2); published on 15 March 2016

1) I was wondering why the authors did not include the topsoil. This is surprising as it was said at several passages in the Introduction and Discussion chapters that for topsoil, aboveground litter plays an important role for SOM. Similarly, a negative point is that several parameters were obtained solely down to a depth of 110 cm.

This study was performed in the framework of a larger research project on subsoil carbon dynamics which uses a comparable sampling scheme (starting in 10 cm) at different sites excluding the A horizons (5 cm). We consider the A horizon of being of minor importance since this study should solely focus on subsoil. Moreover, we needed to sample large and comparable volumes of soil from each sampling depth (using steel cylinders) to obtain enough material for some analysis (e.g. compound-specific 14C), which would not be possible from a 5 cm thick A horizon. Unfortunately, we were able to fractionate and analyse only 5 depth because of limited capacities and funding in our collaborative project.

2) PCA as the only statistical tool is not sufficient for a comprehensive and sound interpretation, especially when taking into account that some of the studied soil properties may correlate with each other. Here, PCA did not yield unambiguous results, so it is not safe to declare roots as the main factor, especially as dissolved organic carbon was not investigated.

In addition to the PCA we insert general linear regression models with 14C as dependent variable in relation to the soil parameters investigated. These results confirm the importance of the root mass already indicated by the PCA. The regression analysis are shown in a new Figure 3 including regression coefficients.

3) It is nowhere explained, why it is desirable to improve understanding of SOM cycling and turnover. One or two sentences would be enough to show the reasons for such research like in the current study.

We added two sentences at the beginning of the introduction explaining the overall importance of the study.

Detailed remarks:

TITLE 4) I miss the information, that the soil under a tree, thus likely forest soil was investigated. This information does not appear before the middle of the Abstract, and

towards the end of the Introduction.

We added this information in the abstract and introduction.

ABSTRACT

5) Lines 28f: This is a rather sudden transition from general points to the current study.

We modified the abstract according to the remark.

6) Lines 32f: It was not mentioned here that only fine roots were analysed.

We added this according to your comment.

7) Line 45: to determine SOM" sounds strange. Please rephrase.

Changed according to your comment

INTRODUCTION

8) I missed some references like Marschner et al. (2008) when talking about different turnover times of SOM compounds (lines 54ff), or Kautz et al. (2013). Although the latter review deals with nutrient uptake from arable and not forest subsoils, it could be included in the current study, as the Introduction is partially written very general.

We are very thankful for the suggested references and added Marschner et al. (2008). In our opinion, the study of Kautz et al. (2013) does not fit completely in this context.

MATERIAL AND METHODS

9) Lines 120ff: If profile A was sampled directly beneath the tree, I am sure that there were also coarse roots present.

Our analysis on root mass was exclusively based on fine roots and is explained in more detail in chapter 2.3.1.

10) Lines 129f: Please give the temperature used for combustion of the samples.

We added the combustion temperature.

11) C2 Lines 161f: The authors treated the samples with acid to remove inorganic carbon. This contradicts the statement made in lines 129f.

Yes, but it is essential to remove any potential inorganic carbon for 14C analysis which may also derive from packing material. AMS 14C analysis are done on $\mu$g to mg amounts of soil and are easily contaminated. We chose a very weak acid treatment with which we did not influence 14C results (according to own tests).

12) Lines 180f: Why were some analyses like particle size not measured on the complete sample set?

According to comment 1, the density and particle fractionation could only performed down to 110 cm depth.

RESULTS

13) Lines 273ff: In my opinion, it is not necessary to give decimals for the pMC values.

We agree in this point, but we will maintain decimals for the pMC values since the measurement errors ranging on this scale.

14) Lines 312ff: The phrasing with the "<" signs is not easy to read.

Changed according to your comment.

DISCUSSION

15) Chapter 4.1 has the heading "Influence of root-derived OC on 14C distribution". However, first it begins with repetition from the Introduction as well as citing references rather than discussing the own data, which I find inappropriate. Second, the complete second paragraph discusses the root depth distribution rather than its influence. The actual statement made about root influence on 14C distribution is restricted to the end of the chapter and is written rather vague.

We agree in this point and rewrote this part.

16) At the end of chapter 4.3 (lines 433f.), it sounds like grain size would have a stronger effect on 14C distribution than roots, which contradicts the overall statement of the study. Please rephrase.

We rephrased this part to make clear that there is only an apparent influence of grain size on 14C distribution.

Further notes:

17) Language is partially bad and has to be considerably improved, also to avoid misunderstandings in case of ambiguous statements (e.g. lines 140ff, lines 190ff.).

The language has been edited.

18) Throughout the manuscript, SI units should be used, also for C contents.

Changed according to your comment

19) From Fig. 1, I can not clearly see if the horizon boundaries were even or undulating, and if they occurred at identical depth throughout the transect. Maybe a picture would be good.

We added the horizon boundaries in this figure to make clear that the samples were taken at identical depths and horizons.

20) I also noticed that the horizon terms refer to the German soil classification system, whereas the soil classification itself refers to WRB. Please use a uniform system.

We changed this according to your comment.

21) The supplement shows the same data like Figure 2. One of the two might be skipped to avoid repetition of the same data.

We skipped Figure 2 and replaced this by the supplementary table which is now table 1.